# The Anti-Arthritic Potential of the Ethanolic Extract of *Salvia Lachnostachys* Benth. Leaves and Icetexane Dinor-Diterpenoid Fruticuline B

**DOI:** 10.3390/ph17091226

**Published:** 2024-09-18

**Authors:** Natália de M. Balsalobre, Elisangela dos Santos-Procopio, Cristhian S. Oliveira, Silvia C. Neves, Maria H. Verdan, Saulo E. Silva-Filho, Rodrigo J. Oliveira, Maria É. A. Stefanello, Cândida A. L. Kassuya

**Affiliations:** 1Faculty of Health Sciences, Federal University of Grande Dourados, Dourados 79804-970, MS, Brazil; nataliabalsalobre@hotmail.com (N.d.M.B.); elisangelaprocopiosan@gmail.com (E.d.S.-P.); 2Chemistry Department, Federal University of Paraná, Curitiba 81530-900, PR, Brazil; cristhian.so@hotmail.com (C.S.O.); stefanello.elida@gmail.com (M.É.A.S.); 3Faculty of Health, Federal University of Mato Grosso do Sul, CeTroGen, Campo Grande 79070-900, MS, Brazil; nevessilva@gmail.com (S.C.N.); rjo.rodrigojulianooliveira@gmail.com (R.J.O.); 4Postgraduate Program in Chemistry, Federal University of Grande Dourados, Dourados 79804-970, MS, Brazil; mhelenaverdan@gmail.com; 5Faculty of Pharmaceutical Sciences, Food and Nutrition, Federal University of Mato Grosso do Sul, Campo Grande 79070-900, MS, Brazil; saulo.e@ufms.br

**Keywords:** articular inflammation, Lamiaceae, nociception, icetexane diterpenoids

## Abstract

The decoction of *Salvia lachnostachys* Benth. leaves is used in Brazilian folk medicine for anti-spasmodic, antipyretic, and anxiolytic purposes. Some of the biological effects of an *S. lachnostachys* extract have been shown to be anti-inflammatory, anti-cancer, and antidepressant effects. In addition, this medicinal plant produces several compounds including icetexane diterpenoids, such as fruticuline A and fruticuline B. The aim of the present work was to evaluate the anti-hyperalgesic and anti-inflammatory properties of fruticuline B (FRUT B) and the ethanolic extract obtained from the leaves of *S. lachnostachys* (EESL) in experimental mouse models. EESL (30, 100, and 300 mg/kg) and FRUT B (1 mg/kg) were evaluated in articular inflammation-induced models in *Swiss* mice. In articular inflammation induced by Zymosan, EESL (300 mg/kg) and FRUT B (1 mg/kg) significantly reduced mechanical hyperalgesia (83.17% inhibition for EESL and 81.19% for FRUT B); edema (68.75% reduction for EESL and 33.66% for FRUT B); leukocyte migration (81.3% for EESSL and 92.2% for FRUT B), and nitric oxide production (88.3% for EESL and 74.4% for FRUT B). The exposure to fruticuline B significantly inhibited the edema (51.5%), mechanical (88.12%) and cold hyperalgesia (80.8%), and myeloperoxidase (MPO) (63.4%) activity 24 h after CFA injection. In the pleurisy model, FRUT B reduced 89.1% of leukocyte migration and 50.3% in nitric oxide production. Four hours after carrageenan injection, FRUT B (1 mg/kg) diminished 89.11% of mechanical hyperalgesia, 65.8% of paw edema, and 82.12% of the response to cold hyperalgesia. In the MTT test, EESL and fruticuline B caused no cytotoxicity. The present study revealed, for the first time, the anti-arthritic and anti-nociceptive effects of FRUT B, pointing out the therapeutic potential of the species to control inflammation and nociception. Future studies are needed to evaluate other biological properties of fruticuline B and to better understand its mechanism of action.

## 1. Introduction

Inflammation is a defense response of the organism triggered by harmful or infectious agents. The inflammatory process can be acute or chronic, and generally, when not properly solved, it can become persistent, leading to an exaggerated response of the immune system [1]. Inflammation, redness, edema, pain, and joint injury [2] may be present in diseases such as rheumatoid arthritis [3]. These clinical signs can be treated with glucocorticoid analogs and non-steroidal anti-inflammatory drugs (NSAIDs). Most drugs cause adverse effects; thus, drug development and new effective treatment alternatives are still needed [4].

The discovery of bioactive substances extracted from plants has a significant impact on drug development, opening new possibilities for therapeutic treatment. The traditional use of medicinal plants has proven to be an effective option for the treatment of inflammatory diseases [5]. Among Brazilian plant species with potential for new drug development, *Salvia lachnostachys* Benth. (Lamiaceae family), an herbaceous plant native to the southern regions of Brazil and commonly known as “melissa”, stands out.

The ethanolic extract of *S. lachnostachys* leaves (EESL) demonstrated an anti-inflammatory effect in pain and inflammation models [6]. Moreover, it has an antitumor and chemopreventive effect against solid Ehrlich carcinoma [7]. Santos Radai and coauthors (2018) demonstrated that the EESL has anti-arthritic activity and no toxic genetic effect, being safe for usage [8]. The EESL additionally exhibited anti-hyperalgesic, antidepressant, and anti-nociceptive effects [9]. This medicinal plant is used in folk medicine as an antipyretic agent, and non-steroidal anti-inflammatory drugs (NSAIDs) are used to treat fever, pain, and inflammation. In this context, the evaluation of an *S. lachnostachys* extract and fruticuline B (FRUT B) in experimental models of inflammation is relevant.

The EESL has been previously characterized for its chemical constituents [7,10,11]. Phytochemical investigation revealed the predominant presence of unique icetexane *nor*- and *dinor*-diterpenoids, along with ursane and oleanane triterpenoids in fractions of low and medium polarity. Additionally, rosmarinic acid, a phenolic compound commonly found in species of the Lamiaceae family, predominates in the more polar fractions [11,12]. Icetaxenes are biosynthesized from abietanes, featuring a tricyclic 6-7-6 structure (systematic name: 9(10/20)-*abeo*-abietane) and containing a *para*-quinone group [13,14]. In particular, the icetexane FRUT B, a *dinor*-diterpenoid that has a rare tricyclic anthracene-based structure, was initially isolated from *S. fruticulosa* [13]. This represents the second recorded isolation of this diterpenoid. The extraction also yielded one of the main chemical constituents, the icetexane fruticuline A (a *nor*-diterpenoid) (Figure 1). Furthermore, other minor icetexanes were isolated, including isofruticuline A, demethylfruticuline A, demethylfruticuline B, 20-hydroxyfruticuline B, and 6-hydroxyisofruticuline A [10,11].

From a pharmacological perspective, fruticuline A has been extensively studied and exhibits significant biological properties, including anti-inflammatory, anti-nociceptive, and antineoplastic activities, as well as antidepressant effects [7,10,13,15]. Similarly, demethylfruticuline A (Figure 1) is known to induce apoptosis [16]. Both fruticuline A and demethylfruticuline A have shown notable binding affinity to CDK-2 compared to known CDK-2 inhibitors [17]. However, FRUT B has been less explored regarding its biological properties, showing only moderate cytotoxic activity (IG_50_ 16.4–116.4 µM) against human tumor cell lines and high antioxidant activity assessed by the ORAC-FL method (1.8 TE) [10,11].

In this perspective, we evaluated the effects of FRUT B in murine models of inflammation, pain, and toxicity, aiming to expand the pharmacological knowledge of these unique chemical skeletons. Additionally, the present study also assessed the effects of the EESL.

## 2. Results

### 2.1. Isolation and Identification of FRUT B

Pure FRUT B was isolated using semi-preparative HPLC and its structure was confirmed by NMR (^1^H, HSQC, and HMBC). The obtained NMR spectra were identical to those previously reported in the literature (Appendix A) [10]. The ¹H NMR showed the typical signals of FRUT B: an isopropyl group (doublet at δ_H_ 1.35, six hydrogens; septet at δ_H_ 3.49, one hydrogen), two tetrasubstituted aromatic rings (two doublets at δ_H_ 7.14 and δ_H_ 7.16, one hydrogen each; and two singlets of two unprotected hydrogens at δ_H_ 8.45 and δ_H_ 8.69, one hydrogen each), a methoxy group (singlet at δ_H_ 3.95, three hydrogens), a methyl group (δ_H_ 2.75, three hydrogens), and a hydroxyl group (broad singlet at δ_H_ 7.55, one hydrogen). Besides the solvent signal (CDCl_3_; δ_H_ 7.26), no other signals were observed, confirming the absence of impurities, such as aliphatic compounds or other aromatic diterpenes, like fruticuline A.

### 2.2. The Oral Exposure to EESL and FRUT B Reduced Zymosan-Induced Knee Inflammation

After 4 h of Zymosan administration in the knee, oral exposure of female mice to a dose of 100 or 300 mg/kg of EESL, FRUT B, and prednisolone (PRED) significantly inhibited mechanical hyperalgesia in 68.19, 75.17, 81.19, and 91.17%, respectively (Figure 2A). The inhibition of mechanical hyperalgesia induced by EESL was 80.21% for 100 mg/kg and 83.17% for 300 mg/kg, while for FRUT B, it was 81.19, and for prednisolone (PRED), it was 85.15% (Figure 1), after 6 h of Zymosan administration in the knee. The dose of 30 mg/kg of EESL did not interfere with mechanical hyperalgesia (Figure 1 and Figure 2A). After 4 h of Zymosan administration in the knee, no doses of EESL, FRUT B, and prednisolone (PRED) groups interfered statistically with knee edema when compared with the control group (Figure 2C). After 6 h of Zymosan administration in the knee, the oral administration of 100 mg/kg of EESL prevented the knee edema in 66.25% while the dose of 300 mg/kg prevented it in 68.75%, FRUT B in 33.66%, and prednisolone (PRED) in 97.3% (Figure 2D).

In relation to the total migration of leukocyte cells to the knee, the doses of 100 and 300 mg/kg of EESL inhibited 67.2 and 81.3% while FRUT B inhibited 92.2% and prednisolone (PRED) inhibited 89.1% (Figure 2E). Regarding nitric oxide production in the knee, 100 and 300 mg/kg doses of EESL inhibited 68.6 and 88.3%, while FRUT B inhibited 74.4%, and prednisolone (PRED) inhibited 85.1% (Figure 2F).

The oral exposure to the doses of 100 and 300 mg/kg of EESL for mechanical hyperalgesia evaluation did not differ between themselves; however, they differed from the control group and from the dose of 30 mg/kg of EESL. The results suggested that EESL acts in a dose-dependent manner since different doses induced distinct responses (Figure 2A,B).

### 2.3. Oral Exposure to FRUT B Reduced CFA-Induced Acute Paw Inflammation

The oral exposure to 1 mg/kg, but not to 0.3 mg/kg, of FRUT B significantly diminished the mechanical hyperalgesia, leukocyte migration, and nitric oxide production 3, 4, and 24 h after CFA injection.

Three hours after the CFA injection, the mechanical hyperalgesia reduction was 94.6% for FRUT B (1 mg/kg), while for prednisolone (PRED), it was 91.9% (Figure 3A). Four hours after the CFA injection, the mechanical hyperalgesia reduction was 86.14% for FRUT B (1 mg/kg), while for prednisolone (PRED), it was 75.3% (Figure 3B). Twenty-four hours after the CFA injection, the mechanical hyperalgesia reduction was 88.12% for FRUT B (1 mg/kg), while for prednisolone (PRED), it was 87.17% (Figure 3C).

Three hours after CFA injection, allodynia reduction to cold was 77.11% for 1 mg/kg of FRUT B, while prednisolone (PRED)’s decrease was 66.12% (Figure 3D). Four hours after CFA injection, allodynia reduction to cold by FRUT B (1 mg/kg) was 64.15%, while prednisolone (PRED) diminished it by 68.12% (Figure 3E). Twenty-four hours after CFA injection, allodynia reduction to cold was 80.8% for FRUT B (1 mg/kg), while prednisolone (PRED) diminished it by 84.7% (Figure 3F).

The reduction in paw edema was 63.3% for FRUT B, while for prednisolone (PRED), it was 70.0%. The inhibition was 63.7% for FRUT B (1 mg/kg) and 84% for prednisolone (PRED) 4 h after CFA. Twenty-four hours after CFA, the inhibition was 51.5% for FRUT B (1 mg/kg) and 80.1% for prednisolone (PRED) (Figure 3G).

The myeloperoxidase activity (measured 24 h after CFA injection) was 63.4% inhibited by FRUT B (1 mg/kg), while it was 76.9% inhibited by the prednisolone (PRED) group (Figure 3H).

### 2.4. The Oral Exposure to FRUT B Diminished Carrageenan-Induced Pleurisy

The evaluation of pleurisy induced by carrageenan showed a reduction in leukocyte migration of 89.1% for FRUT B (1 mg/kg), and for prednisolone (PRED), the reduction was 95.0% compared to the control group. There was a decrease of 50.3% in nitric oxide production for FRUT B (1 mg/kg), and for prednisolone (PRED), the reduction was 52.3%. The oral exposure of FRUT B and prednisolone (PRED) did not differ between themselves; however, they differed from the control group in relation to leukocyte migration or nitric oxide production (Figure 4A,B).

### 2.5. The Oral Exposure to FRUT B Decreased Carrageenan-Induced Paw Inflammation

The oral exposure to FRUT B and prednisolone (PRED) diminished the mechanical hyperalgesia, paw edema, and response to cold 3 and 4 h after carrageenan injection. After three hours of carrageenan, the mechanical hyperalgesia was decreased by FRUT B in 77.0% (0.3 mg/kg) and in 82.0% (1 mg/kg) and by prednisolone (PRED) in 93.7% (3 mg/kg) (Figure 5A). After 4 h of carrageenan injection, the mechanical hyperalgesia was inhibited by FRUT B in 85.15% (0.3 mg/kg) and in 89.11% (1 mg/kg), and by prednisolone (PRED) in 93.7% (3 mg/kg) (Figure 5B).

After 3 h of carrageenan, the cold allodynia was inhibited by FRUT B in 71.21% (0.3 mg/kg) and in 79.9% (1 mg/kg), and by prednisolone (PRED) in 86.9% (3 mg/kg) (Figure 5C). After 4 h of carrageenan, the cold allodynia was inhibited by FRUT B in 88.1% (0.3 mg/kg) and in 82.12% (1 mg/kg), and by prednisolone (PRED) in 88.7% (3 mg/kg) (Figure 5D).

After 3 h of the stimulus, the edema was inhibited by FRUT B in 73.8% (0.3 mg/kg) and in 82.5% (1 mg/kg), and by prednisolone (PRED) in 64.3% (3 mg/kg) (Figure 5E). After four hours of the stimulus, the edema was inhibited by FRUT B in 68.7% (0.3 mg/kg) and in 65.8% (1 mg/kg), and by prednisolone (PRED) in 73.3% (3 mg/kg) (Figure 5F).

### 2.6. FRUT B Did Not Present In Vitro Cellular Toxicity in the MTT Test and Did Not Present Alterations in Parameters of the Acute Oral Toxicity Test

In the MTT test, FRUT B at concentrations of 3, 10, 30, and 90 µg/mL presented cell viability of 82.0%, 89.3%, 90.9%, and 83.5%, respectively, indicating that this compound did not induce cellular toxicity in any tested concentrations.

The single oral exposure to EESL and FRUT B (doses of 500, 1000, and 2000 mg/kg) did not induce death and did not indicate alterations in the behavioral parameters of the acute oral toxicity test. The body weight, the amount of food consumption, and the weight of organs observed in FRUT B groups did not show significant differences when compared to the control group.

## 3. Discussion

FRUT B is an aromatic diterpene isolated from *S. lachnostachys* leaves and the results herein show, for the first time, the anti-inflammatory effects of this compound. Another compound found in *S. lachnostachys* is fruticuline A; however, the anti-inflammatory potential of fruticuline A, as well as the ethanolic extract of *S. lachnostachys* (EESL), was previously explored in animal models of inflammation. The oral administration of EESL, in a dose-dependent manner, inhibited parameters of inflammation [6,7,9]. Few studies have evaluated the anti-inflammatory effect and therapeutic potential of FRUT B, and it is important that the scientific investigation of FRUT B in relation to toxicity and pharmacology is carried out. The dose of 1 mg/kg of FRUT B was used to perform the biological evaluation since the yield of FRUT B in EESL was around 0.7%, and the effective dosage was 100 mg/kg. In the present study, FRUT B (1 mg/kg) also effectively inhibited the inflammatory parameters induced by Zymosan, CFA, and carrageenan.

The Zymosan articular inflammation model is used to analyze the effects of substances on parameters such as knee edema formation, mechanical hyperalgesia, and leukocyte migration. The oral administration of 100 and 300 mg/kg of EESL significantly reduced, in a dose-dependent manner, the mechanical hyperalgesia and decreased other Zymosan parameters analyzed. The EESL treatments (dose of 300 mg/kg) blocked the mechanical hyperalgesia 4 and 6 h after the intra-articular injection of Zymosan (Figure 1). Inflammatory mediators such as cytokines (IL-1 beta and IL-6), TNF, and endothelins are increased in the knee when animals are analyzed at 6 h after Zymosan knee injection in mice [18,19]. The oral administration of 300 mg/kg EESL, but not fruticuline B, was effective in inhibiting knee edema at 6 h (not 4 h), suggesting that the mechanism of action may involve mediators that increase at this time point (Figure 2C,D). In relation to edema, other compounds present in EESL may be involved in the mechanism of action of reduction in edema in this model. The inhibitory efficacy of EESL on leukocyte migration was higher than prednisolone (PRED) (Figure 1). FRUT B (1 mg/kg) exhibited efficacy against mechanical hyperalgesia and inhibited leukocyte migration, showing anti-inflammatory properties. The present study was the first to show that EESL and FRUT B present anti-arthritic potential against Zymosan articular inflammation.

The inflammatory response induced by CFA in mice is another inflammatory model used to efficiently evaluate new products. The hyperalgesia, edema, and leukocyte migration to the paw are inflammatory parameters induced by CFA [18,20,21], and the oral administration of 1 mg/kg of FRUT B was able to inhibit all these parameters. The oral administration of EESL (100 mg/kg) for 21 days to mice showed anti-hyperalgesic and anti-edematogenic properties in the CFA model of inflammation [8]. The present study also verified the anti-arthritic potential of FRUT B in CFA-induced acute paw inflammation.

The EESL showed anti-inflammatory efficacy in the pleurisy model induced by carrageenan (Figure 4). Models of pleurisy induced by carrageenan are important to study the influence of new products on phenomena such as migrating cells and exudate formation [22]. In the pleurisy model, the literature showed that the highest neutrophil accumulation was observed 4 h after the intrapleural injection of carrageenan [23]. The doses of 1 mg/kg of FRUT B caused a significant inhibition of leukocyte migration and nitric oxide production induced by carrageenan administration into the pleural exudate (Figure 4A,B), showing the anti-inflammatory properties of FRUT B. EESL and fruticuline A exhibited anti-hyperalgesic, antidepressant, and anti-nociceptive effects in the nerve injury (SNI) in rats and nociceptive behavior induced by formalin [9]. EESL and fruticuline A display anti-inflammatory and analgesic properties in paw edema and pleurisy induced by carrageenan injection [6]. FRUT B was also analyzed in the carrageenan-induced paw inflammation model and the parameters analyzed were mechanical hyperalgesia, cold allodynia, and edema (Figure 5). The oral exposure to FRUT B reduced all analyzed parameters (Figure 5). These results showed for the first time the potential of FRUT B as an anti-inflammatory agent and showed the inhibition of mechanical hyperalgesia, cold response, edema, leukocyte migration, and nitric oxide induced by carrageenan.

The safe oral administration of EESL and fruticuline A was shown by Santos Radai and coworkers (2018) [8]. Fruticuline A and B are secondary metabolites with a diterpene skeleton, so far only found in species from the *Salvia* genus. Fruticuline A has shown to be promising for cancer treatment because it causes cellular death in vitro, and in in vivo assays, the cells suffered necrosis [13].

This study has some limitations; for example, because of the small quantity of fruticuline B, it was not possible to test this substance in various animal toxicity models. Future studies are needed to evaluate other properties of fruticuline B and to better understand its mechanism of action. Another limitation is the use of only one genus of mice and only a few doses of the extract and isolated compound in the experimental models, but the ethics of the use of animal experimentation restrict the use of animals.

Two studies [7,10] showed that fruticuline A and isofruticuline A had the highest cytotoxic activity against several cancer lines; however, fruticuline B had low cytotoxicity against all cell lines tested [7]. Therefore, fruticuline B was evaluated in the MTT assay. The MTT assay also showed that FRUT B has no cellular toxicity. These preliminary evaluations showed, for the first time, that FRUT B did not show potential toxicity. No toxicity was observed in the acute toxicity assay when animals were treated with an ethanolic extract of *S. lachnostachys* leaves [7]; however, this plant has been described as a cytotoxic agent in the literature. According to Santos Radai et al. [8], the ethanolic extract of this plant has no genotoxic effect in male Swiss mice treated at a dose of up to 1000 mg/kg.

## 4. Materials and Methods

### 4.1. Plant Material

Leaves of *Salvia lachnostachys* Benth. were collected and identified in Curitiba, Paraná State, Brazil (25°30′44.6″ S, 48°18′7.13″ W), in May 2010 by Élide P. Santos. A voucher specimen was deposited in the Herbarium of the Federal University of Paraná (UPCB 85285). This study of this species was registered in the National System for the Management of Genetic Heritage and Associated Traditional Knowledge (SISGEN) under the code A19F875.

### 4.2. Extract Preparation and Isolation of FRUT B

Leaves of *Salvia lachnostachys* (415.3 g) were dried and powdered, and subsequently, this material was submitted to a successive extraction process with hexanes (Hex, mixture of isomers) and EtOH (2.0 L, three times each solvent at room temperature). The solvents were removed under reduced pressure, yielding extracts of Hex (11.3 g) and EtOH (EESL, 47.6 g). An aliquot (2.6 g) of the EESL was reserved, and the remaining (45.0 g) was submitted to vacuum liquid chromatography, eluted with Hex, yielding a fraction (A, 1.8 g) after the solvent removal. This fraction was fractionated by CC chromatography (silica gel), and eluted with several solvents in polarity order (Hex, Hex-CH_2_Cl_2_ (1:1, *v*/*v*), CH_2_Cl_2_–acetone (from 1:0 to 1:1, *v*/*v*), and finally with acetone), resulting in 14 subfractions (A1–A14), after a TLC analysis. Subfraction A6 (478.4 mg) was recrystallized in Hex, yielding a mixture of fruticuline A and FRUT B. An aliquot of this mixture (107.5 mg) was purified by semi-preparative HPLC (a Waters apparatus equipped with a PDA detector and a semi-preparative Nucleosil 100-5 C18 column, 250 × 10 mm), using MeOH-MeCN (50:50, *v*/*v*), isocratic mode (20 min), to obtain fruticuline A (58.8 mg, retention time = 9.2 min) and FRUT B (21.8 mg, retention time = 10.3 min). Compounds were identified by the comparison of their NMR data (Bruker spectrometer Avance 400, using deuterated chloroform, Bruker, Karlsruhe, Germany) with the literature [10].

### 4.3. Animals

The experiments were performed with female and male Swiss mice (aged between 8 and 10 weeks; males weighing from 33 to 39 g, females weighing from 28 to 32 g). The animals received food and water ad libitum. They were kept in an environment with a controlled temperature of 22 ± 2 °C, and 12 h of a light/dark cycle. The experiments were performed in the Pharmacology Laboratory of the Faculty of Health Sciences (FCS), after the approval by the Ethics Committee on Animal Use (CEUA) of UFGD under protocol CEUA 22/2022.

### 4.4. Zymosan-Induced Knee Inflammation

The animals (female mice) were randomly distributed in groups and were treated with a vehicle (saline solution (0.9%); negative control group), EESL (30, 100, and 300 mg/kg), FRUT B (1 mg/kg), and prednisolone (PRED, 3 mg/kg, positive group) one hour before Zymosan injection. The inflammation in the knee was induced by an intra-articular injection of Zymosan (500 µg per cavity with 25 µL saline) [17,24]. The mechanical hyperalgesia, cold response, and edema were measured 4 and 6 h after Zymosan injection. The animals were euthanized (ketamine—300 mg/kg, intraperitoneal (i.p.) + xylazine—30 mg/kg, i.p.) 6 h after Zymosan injection. Subsequently, the synovial cavities of the knees were washed with a solution of PBS/EDTA, and the synovial liquid was collected by aspiration 6 h after Zymosan injection. Total cell counts and NO were measured using a Neubauer chamber and Griess methods, respectively.

### 4.5. CFA-Induced Acute Paw Inflammation

In previous work of our research group, mice were subjected to prolonged treatment with an ethanolic extract of *S. lachnostachys* Benth. leaves and this extract exhibited anti-inflammatory parameters in the Complete Freund’s Adjuvant (CFA) model [8]. For the study, the animals (male mice) were randomly distributed in groups and were treated twice (one hour before and 24 h after CFA injection) with a vehicle (saline solution, 0.9%; negative control group), FRUT B (0.3 and 1 mg/kg), and prednisolone (PRED, 3 mg/kg, positive control group). At time zero, 20 µL of CFA (suspension containing dead *Mycobacterium tuberculosis*, paraffin (85%), and monooleate (15%) oils) was injected into the right hind paw. The mechanical hyperalgesia [25], cold response [18], and edema [26] were measured at 3, 4, and 24 h after CFA injection. The animals were euthanized (ketamine—300 mg/kg, intraperitoneal (i.p.) + xylazine—30 mg/kg, i.p.) 6 h after Zymosan injection. Subsequently, the skin of the paw of each animal was removed for measurements of MPO and NAG activities.

### 4.6. Carrageenan-Induced Acute Pleurisy

One hour prior to carrageenan injection, Swiss female mice were distributed into three groups and treated orally with 0.9% saline solution (naive and negative control groups (p.o.)), prednisolone (PRED, 3 mg/kg, positive control group), and FRUT B (0.3 and 1 mg/kg). These groups received 100 μL of a solution of 1% carrageenan while the naive group received a similar volume of the saline solution by the intrapleural route. After 4 h, the animals were euthanized (ketamine—300 mg/kg, intraperitoneal (i.p.) + xylazine—30 mg/kg, i.p.), and the pleural cavity was washed with PBS/EDTA. The total leukocyte number and the level of protein were measured in the pleural exudate.

### 4.7. Cell Viability Assay by Colorimetric Method (MTT)

The cellular cytotoxicity test was performed using a colorimetric method with MTT (Sigma, São Paulo, Brazil) (Figure S4) [20]. The leukocytes were obtained from peripheral blood. The cells were plated with 5 × 10^5^ cells/well (in 96-well plates) in a volume of 100 µL RPMI medium (supplemented with 10% fetal bovine serum (FBS) and penicillin at 100 U/mL + streptomycin at 100 mg/mL). After 90 min of exposure to FRUT B (3, 10, 30, 90 μg/mL), 10 µL of the MTT solution was added (5 mg/mL, Sigma) to each well. After 2 h of incubation at 37 °C, the supernatant was removed and 100 µL of the lysis solution was added to each well. The cells were incubated at 25 °C for 10 min, and subsequently, the absorbance was measured using a microplate reader at 540 nm. The values of the blank wells were discounted, and viability calculations were performed according to Silva-Filho et al. [22].

### 4.8. Statistical Analysis

Data are expressed as the mean ± SEM for each experimental group. The results were statistically analyzed by using a one-way variance analysis (ANOVA), followed by Tukey’s post hoc test. The percentage of inhibition was calculated in relation to the control group prednisolone (PRED). Differences were considered significant when *p* < 0.05.

## 5. Conclusions

This study has demonstrated that the EESL and FRUT B reduced inflammation and hyperalgesia in Zymosan articular pain. Our study was the first to reveal the anti-hyperalgesic, anti-arthritic, and anti-inflammatory potential of FRUT B in experimental in vivo models. In the MTT test, no cytotoxicity was detected by the compound exposure.

## Figures and Tables

**Figure 1 pharmaceuticals-17-01226-f001:**
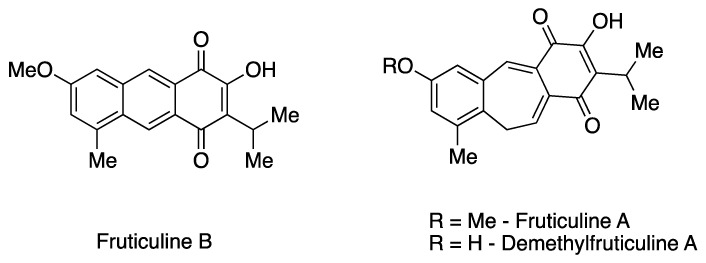
Chemical structures of the icetexane *nor*-diterpenoids fruticuline A and demethylfruticuline A and *dinor*-diterpenoid fruticuline B.

**Figure 2 pharmaceuticals-17-01226-f002:**
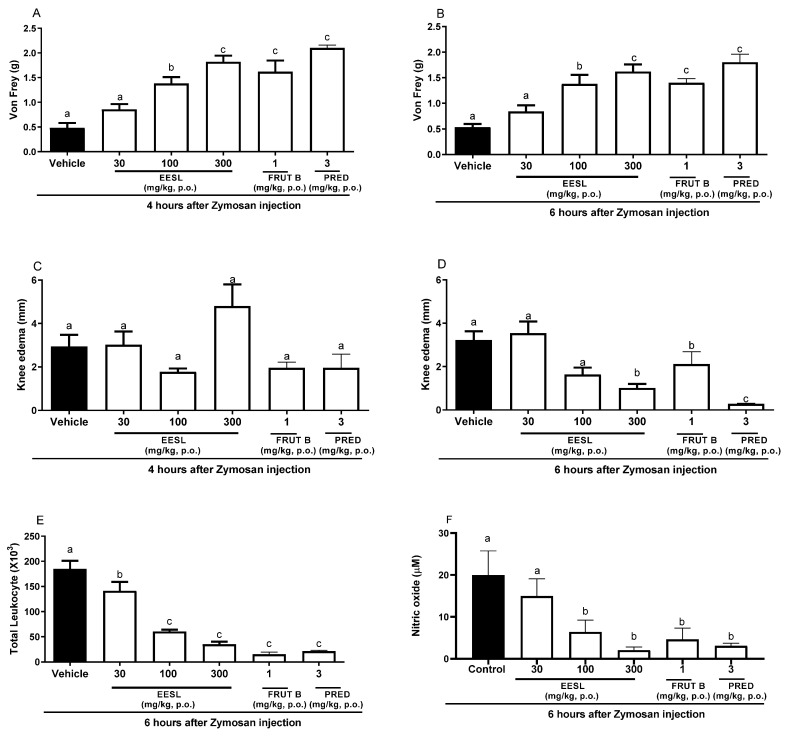
The effect of oral administration of EESL and FRUT B on the Zymosan-induced knee edema, mechanical hyperalgesia, leukocyte migration, and nitric oxide in synovial exudate in mice. The animals received EESL (30, 100, and 300 mg/kg, p.o.), FRUT B (1 mg/kg), vehicle (control), or prednisolone (PRED, 3 mg/kg, p.o.); and 1 h later, an intraplantar injection of Zymosan was administered. (**A**,**B**) represent the evaluation of the mechanical hyperalgesia at 4 and 6 h, respectively, after Zymosan injection. (**C**,**D**) represent the evaluation of the knee edema at 4 and 6 h, respectively, after Zymosan injection; graph (**E**) represents the evaluation of the total leukocyte counts, and (**F**) represents nitric oxide measurements after 6 h after the stimulus. The letters “a”, “b”, and “c” indicate significant differences among groups according to Tukey’s multiple comparison test.

**Figure 3 pharmaceuticals-17-01226-f003:**
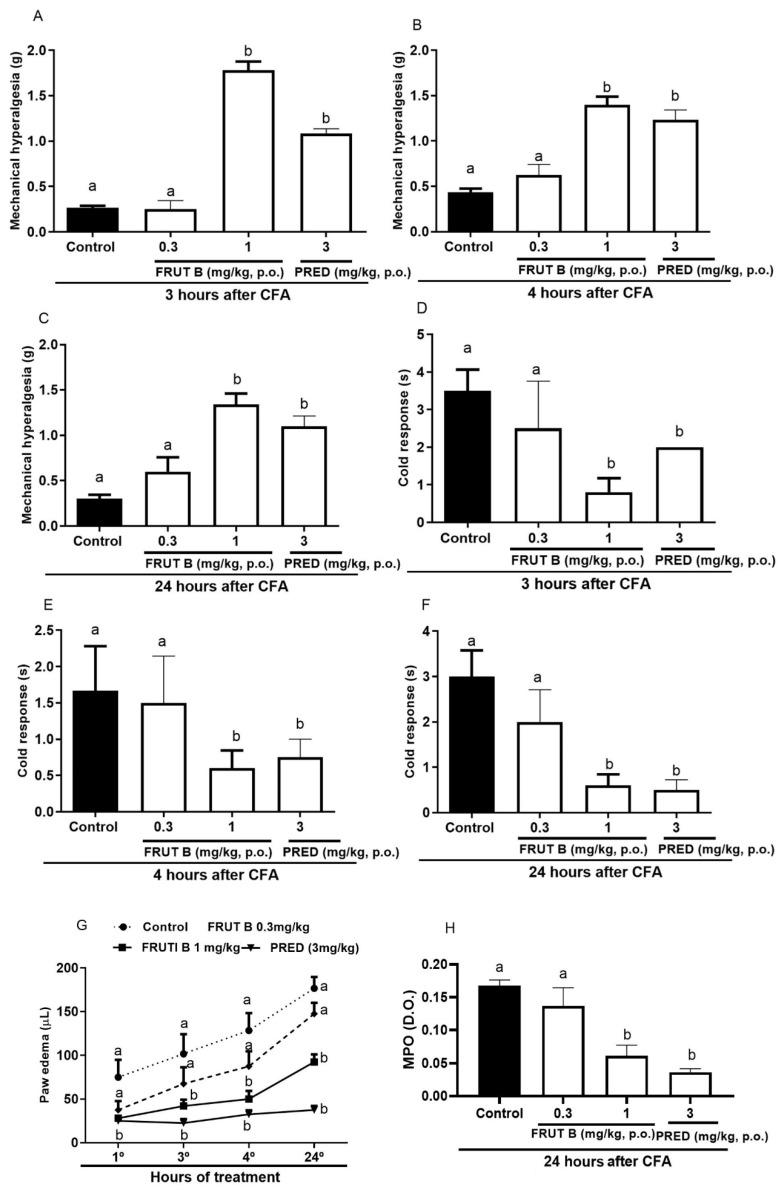
The effect of oral administration of FRUT B on the CFA-induced mechanical hyperalgesia, cold response, paw edema, and increase in myeloperoxidase in mice. The animals received FRUT B (0.3 and 1 mg/kg, p.o.), vehicle (control), or prednisolone (PRED, 3 mg/kg, s.c.), and 1 h later, an intraplantar injection of CFA was administered. (**A**–**C**) represent the evaluation of the mechanical hyperalgesia 3, 4, and 24 h, respectively, after CFA injection. (**D**–**F**) represent the analysis of the cold allodynia at 3, 4, and 24 h, respectively, after CFA injection. (**G**) represents the analysis of paw edema at 3, 4, and 24 h and (**H**) MPO activity measurements 24 h after the stimulus. The letters “a”, “b”, and “c” indicate significant differences among groups according to Tukey´s multiple comparison test.

**Figure 4 pharmaceuticals-17-01226-f004:**
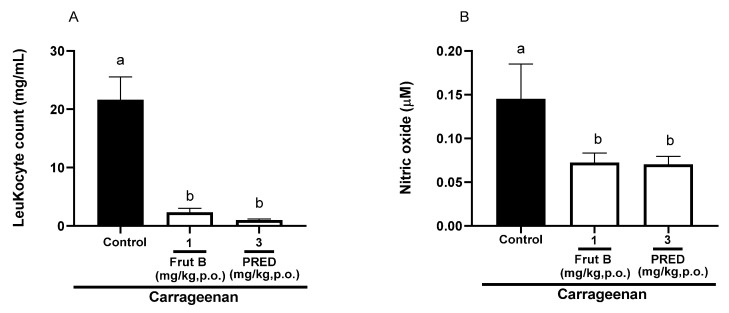
The effect of oral administration of FRUT B on the carrageenan-induced leukocyte migration and nitric oxide levels in pleura in mice. The animals received FRUT B (1 mg/kg, p.o.), vehicle (control), or PRED (3 mg/kg, p.o.), and 1h later, an intraplantar injection of carrageenan was administered. (**A**) represents the evaluation of leukocyte migration and (**B**) represents the evaluation of the nitric oxide levels in pleural exudate 4 h after carrageenan injection. The letters “a” and “b” indicate significant differences among groups according to Tukey’s multiple comparison test.

**Figure 5 pharmaceuticals-17-01226-f005:**
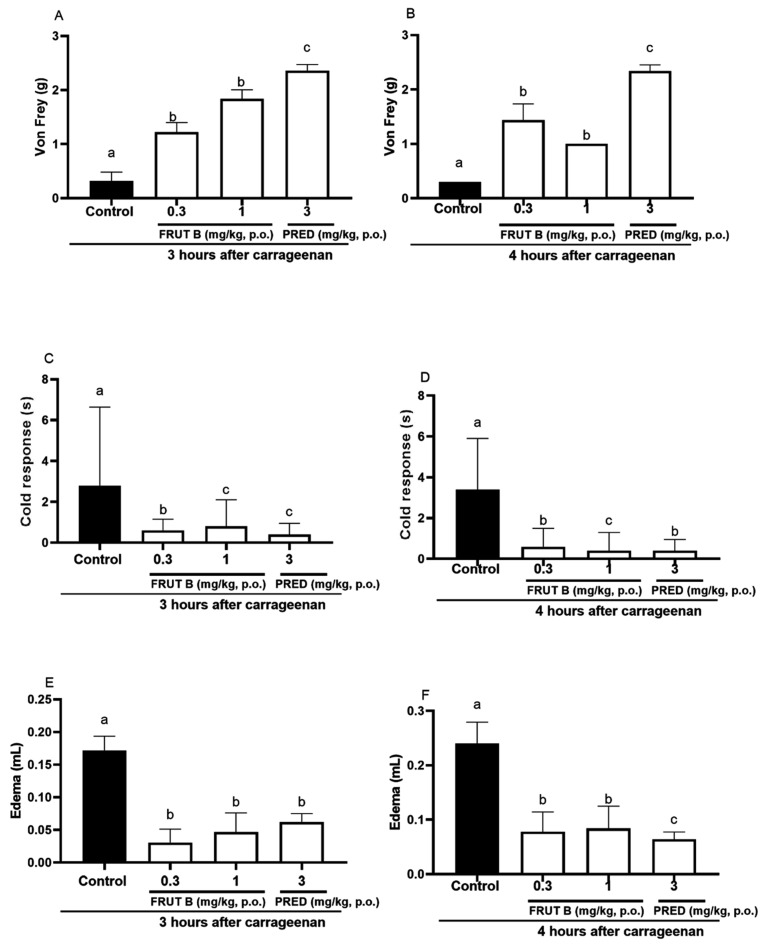
The effect of oral administration of FRUT B on the carrageenan-induced paw edema, mechanical hyperalgesia, and cold response. The animals received FRUT B (0.3 and 1 mg/kg, p.o.), vehicle (control), or prednisolone (PRED, 3 mg/kg, p.o.), and 1 h later, an intraplantar injection of carrageenan was administered. (**A**,**B**) represent the mechanical hyperalgesia 3 and 4 h, respectively, after carrageenan injection. (**C**,**D**) represent the cold allodynia at 3 and 4 h after carrageenan injection. (**E**,**F**) show the inhibition of edema at 3 and 4 h after carrageenan injection. The letters “a”, “b”, and “c” indicate significant differences among groups according to Tukey´s multiple comparison test.

## Data Availability

Data is contained within the article and Appendix A.

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
