# Peer review of "The Anti-Arthritic Potential of the Ethanolic Extract of Salvia Lachnostachys Benth. Leaves and Icetexane Dinor-Diterpenoid Fruticuline B"

_pharmaceuticals, 2024, doi:10.3390/ph17091226_

Round 1
Reviewer 1 Report
Comments and Suggestions for Authors
Overall, while the article conducts several important assays, there are some issues that could still be improved. Here are some of my comments:
Nowadays, conducting assays with only one sex of rodents can lead to misinterpretation of the results, projecting expectations onto organisms that function differently.
The study also seems to lack a clear justification for the use of a particular compound. Is this compound overwhelmingly dominant in the sample? Why focus on this compound specifically? Is it because previous studies have already reached conclusions with Frut A?
The results and discussion sections appear to be limited to a description of the graphs without leading to hypotheses about what might have occurred. Additionally, they do not delve into explaining the importance of each assay and its outcome, leaving some loose ends.
One of my main objections to the methodology is the use of a single individual for the three doses used of the acute toxicology test, while five individuals were used in the control group. It seems the protocol for this methodology was misunderstood, and if so, the assay is invalid.

Author Response
|
Response to Reviewer 1 X Comments
|
||
|
1. Summary |
|
|
|
Thank you very much for taking the time to review this manuscript. Please find the detailed responses below and the corresponding revisions/corrections in track changes in the re-submitted files. |
||
|
2. Point-by-point response to Comments and Suggestions for Authors |
||
|
Comments 1: Overall, while the article conducts several important assays, there are some issues that could still be improved. Here are some of my comments: Nowadays, conducting assays with only one sex of rodents can lead to misinterpretation of the results, projecting expectations onto organisms that function differently. |
||
|
Response 1: Please check that we include these points in the discussion section of the new version of the manuscript.
|
||
|
Comments 2: The study also seems to lack a clear justification for the use of a particular compound. Is this compound overwhelmingly dominant in the sample? Why focus on this compound specifically? Is it because previous studies have already reached conclusions with Frut A? |
||
|
Response 2: This manuscript presents the investigation of fruticuline B, an icetexane with a unique and rare chemical skeleton found in the leaves of Salvia lachnostachys. According to Rodríguez-Hahn (1989), this compound, which has an anthracene core, is biosynthesized through an enzymatic rearrangement of the icetexane skeleton, specifically from fruticuline A. The structural similarity between fruticuline A and B, along with the numerous reports of the biological activity of fruticuline A, justifies the pharmacological approach taken in this study, given the limited biological research on fruticuline B and the need to expand knowledge about this compound. Previous studies have shown that fruticuline A and B are the main diterpenes in the ethanolic extract of Salvia lachnostachys leaves, as evidenced by a chromatographic analysis published earlier (Corso et al., 2019). Given the various biological activities of fruticulin A, we chose to investigate fruticuline B as well, further highlighting the importance of this study.
Rodríguez-Hahn, L.; Esquivel, B.; Sánchez, C.; Estebanes, L.; Cárdenas, J.; Soriano-García, M.; Ramamoorthy, R.T.T.P. Abietane type diterpenoids from Salvia fruticulosa. A revision of the structure of fruticulin B. Phytochem. 1989, 28, 567-570.
Corso, C.R., Stipp, M.C.; Adami, E.R.; da Silva, L.M.; Mariott, M.; de Andrade, S.F.; Ramos, E.A.S.; Klassen, G.; Beltrame, O.C.; Queiroz-Telles, J.E.; de Oliveira, C.S.; Stefanello, M.E.A.; Acco, A. Salvia lachnostachys Benth has antitumor and chemopreventive effects against solid Ehrlich carcinoma. Mol. Biol. Rep. 2019, 46, 4827–4841. https://doi.org/10.1155/2014/835914
|
||
|
Comments 3: The results and discussion sections appear to be limited to a description of the graphs without leading to hypotheses about what might have occurred. Additionally, they do not delve into explaining the importance of each assay and its outcome, leaving some loose ends.
|
||
|
Response 3: Please check that we include these points in the discussion section of the new version of the manuscript.
|
||
|
Comments 4: One of my main objections to the methodology is the use of a single individual for the three doses used of the acute toxicology test, while five individuals were used in the control group. It seems the protocol for this methodology was misunderstood, and if so, the assay is invalid. |
||
|
Response 4: We are in accordance with the comments of the editor and reviewer 1 about this test; please verify that this methodology and results were removed in the new version of the manuscript.
|
||
|
Comments 5 (consideration in pdf of manuscript): Line 21: Rephrase…It seems disconnected from the previous phrase: |
||
|
Response 5: Please, verify that the phrase was altered in the new version of the manuscript.
|
||
|
Comments 6 (consideration in pdf of manuscript): Line 23: The aim of the manuscript seems disconnected with the traditional use. Why did you performed anti-inflammatory experiments with a plants that has other traditional uses? |
||
|
Response 6: Please note that the following sentence was inserted in the introduction section: “This medicinal plant is used in folk medicine as an antipyretic agent, and non-steroidal anti-inflammatory drugs (NSAIDs) are used to treat fever, pain, and inflammation. In this context, the evaluation of S. lachnostachys extract and fruticulin B (FRUT B) in experimental models of inflammation is relevant.”
|
||
|
Comments 7 (consideration in pdf of manuscript): Line 52: This could be increased at the abstract section. |
||
|
Response 7: Please, verify that the sentence “Some of the biological effects of S. lachnostachys extract have been shown to be anti-inflammatory, anti-cancer, and anti-depressant.” was inserted in the abstract section of the new version of manuscript.
|
||
|
Comments 8 (consideration in pdf of manuscript): Line 93: Please add the carbon and HMBC or other spectra of Frut B |
||
|
Response 8:The 13C NMR spectrum was not performed in this study. Since we are dealing with a known substance, as is the case of fruticuline B, where all spectral data have already been described, there is no need to acquire these data again. However, when a 13C spectrum coupled with the hydrogen nucleus (1H) is carried out, as in the case of HSQC and HMBC, the 13C chemical shift data are obtained, as well as the 1H-13C correlations, allowing the comparison of the NMR data with the literature, confirming that our data are identical to those published for fruticuline B. The 1H, HSQC, and HMBC spectra are presented in the Supplementary material..
|
||
|
Comments 9 (consideration in pdf of manuscript): Line 128: Include on the discussion, a possible explanation about why 4 hours after injection the 300 mg/kg dose presented this activity…What would happen to have a dose-dependent manner activity after 6 hours? |
||
|
Response 9: Please, verify that this point was inserted in the conclusion section of the new version of the manuscript.
|
||
|
Comments 10 (consideration in pdf of manuscript): Line 128: Put the 4 hours and 6 hours graphic on the same scale |
||
|
Response 10: Please, verify that this point was inserted in the results section of the new version of the manuscript.
|
||
|
Comments 11 (consideration in pdf of manuscript): Line 208: Why MTT assays was not performed on the EESL? |
||
|
Response 11: Please note that the following sentence has been added to the Discussion section: “Two studies [7, 10] showed that fruticuline A and isofruticuline A had the highest cytotoxic activity against several cancer lines; however, fruticuline B had low cytotoxicity against all cell lines tested [7]. Therefore, Fruticuline B was evaluated in the MTT assay” “No toxicity was observed in the acute toxicity assay when animals were treated with ethanolic extract of S. lachnostachys leaves [7], however, this plant has been described as a cytotoxic agent in the literature. According to Santos Radai et al. [8], the ethanolic extract of S. lachnostachys has no genotoxic effect in male Swiss mice treated at a dose of up to 1000 mg/kg.”
|
||
|
Comments 12 (consideration in pdf of manuscript): Line 212: Include the results on the Supplementary material |
||
|
Response 12: Please verify that the results were not inserted in supplementary material since published articles (for exempla Bernal et al., 2023) of our group accepted the description of percentage values. The values expressed in percentage is commonly described in the literature for this test. Bernal LPT, Leitão MM, Radai JAS, Cardoso CAL, Lencina JDS, Fraga TL, Arena AC, Silva-Filho SE, Kassuya CAL. Analgesic and anti-inflammatory potential of ethanolic extract from Serjania erecta leaves. J Ethnopharmacol. 2023 Mar 1;303:116019. doi: 10.1016/j.jep.2022.116019. Epub 2022 Dec 7. PMID: 36493996.
|
||
|
Comments 13 (consideration in pdf of manuscript): Line 217: According to the description of the Material and Methods section, there was a misunderstood of the protocol. Read and correct that, and if case that the protocol was correctly applied, show this results on the supplementary material |
||
|
Response 13: We are in accordance to the comments of editor and reviewer 1 about this test; please verify that this methodology and results were removed in the new version of the manuscript.
|
||
|
Comments 14 (consideration in pdf of manuscript): Line 219, line 268, 279, line 285 and line 327: Italic |
||
|
Response 14: The italic form was inserted.
|
||
|
Comments 15 (consideration in pdf of manuscript): Line 226: According to the introduction, this is one of the main components of the extract. Is it a major component? Is there some differences with the FRUT A results of previous work? |
||
|
Response 15: Fruticuline B is an icetexane with a unique and rare chemical skeleton found in the leaves of Salvia lachnostachys. According to Rodríguez-Hahn (1989), this compound, which has an anthracene core, is biosynthesized through an enzymatic rearrangement of the icetexane skeleton, specifically from fruticuline A. The structural similarity between fruticuline A and B, along with the numerous reports of the biological activity of fruticuline A, justifies the pharmacological approach taken in this study, given the limited biological research on fruticuline B and the need to expand knowledge about this compound. Previous studies have shown that fruticuline A and B are the main diterpenes in the ethanolic extract of Salvia lachnostachys leaves, as evidenced by a chromatographic analysis published earlier (Corso et al., 2019). Given the various biological activities of fruticuline A, we chose to investigate fruticuline B as well, further highlighting the importance of this study.
Rodríguez-Hahn, L.; Esquivel, B.; Sánchez, C.; Estebanes, L.; Cárdenas, J.; Soriano-García, M.; Ramamoorthy, R.T.T.P. Abietane type diterpenoids from Salvia fruticulosa. A revision of the structure of fruticulin B. Phytochem. 1989, 28, 567-570.
Corso, C.R., Stipp, M.C.; Adami, E.R.; da Silva, L.M.; Mariott, M.; de Andrade, S.F.; Ramos, E.A.S.; Klassen, G.; Beltrame, O.C.; Queiroz-Telles, J.E.; de Oliveira, C.S.; Stefanello, M.E.A.; Acco, A. Salvia lachnostachys Benth has antitumor and chemopreventive effects against solid Ehrlich carcinoma. Mol. Biol. Rep. 2019, 46, 4827–4841. https://doi.org/10.1155/2014/835914
|
||
|
Comments 16 (consideration in pdf of manuscript): Line 238: I would like a more in-depth discussion on the interpretation of each graph. Although is asserted that there was blockade of inflammatory activity, the response to edema does not appear to be significant in the edema graphs for the treatments of interest. If the examination pf the stages of inflammation and the implications of each graph, it would be clearer to understand |
||
|
Response 16: As requested, a discussion about the effects of the extract on edema was inserted in the discussion section.
|
||
|
Comments 17 (consideration in pdf of manuscript): Line 244: In the methodology section, as you will see later, I questioned the lack of continuation with the extract trials. Only now, in the discussion, I could clearly perceive that there are already studies on this; however, I do not feel there is a deep discussion about what might be happening and a potential comparison between the extract and the isolated compound. Overall, I believe this section still needs further development. There are many results and toll available to formulate some hypotheses about what is being observed. |
||
|
Response 17: As requested, a discussion about the limitations and the future perspectives was inserted in the discussion section.
|
||
|
Comments 18 (consideration in pdf of manuscript): Line 288: Removed this phase. The next one explained the obtained extracts |
||
|
Response 18: The phrase was rewritten.
|
||
|
Comments 19 (consideration in pdf of manuscript): Line 312: Why the use of just female mice? |
||
|
Response 19: This experimental model aims to reproduce the clinical parameters of rheumatoid arthritis (RA). As with many autoimmune diseases, RA is more prevalent in women than in men, with an approximate ratio of three to one. The differences in the male and female immune systems are influenced by hormonal distribution, the presence of two X chromosomes instead of a single one, and the distinct response to environmental factors. Several genes located on the X chromosome regulate innate and adaptive immune responses. In addition, sex hormones act as immunoregulators, participating in the secretion of cytokines and chemokines, interacting with inflammatory mediators, and playing a crucial role in the pathobiological differences associated with the disease.
Angum F, Khan T, Kaler J, Siddiqui L, Hussain A. The Prevalence of Autoimmune Disorders in Women: A Narrative Review. Cureus. 2020 May 13;12(5):e8094. doi: 10.7759/cureus.8094. PMID: 32542149; PMCID: PMC7292717. Ortona E, Pierdominici M, Maselli A, Veroni C, Aloisi F, Shoenfeld Y. Sex-based differences in autoimmune diseases. Ann Ist Super Sanita. 2016 Apr-Jun;52(2):205-12. doi: 10.4415/ANN_16_02_12. PMID: 27364395. Klein SL, Flanagan KL. Sex differences in immune responses. Nat Rev Immunol. 2016 Oct;16(10):626-38. doi: 10.1038/nri.2016.90. Epub 2016 Aug 22. PMID: 27546235. Morgacheva O, Furst DE. Women are from venus, men are from Mars: do gender differences also apply to rheumatoid arthritis activity and treatment responses? J Clin Rheumatol. 2012 Aug;18(5):259-60. doi: 10.1097/RHU.0b013e31825833e0. PMID: 22832285.
|
||
|
Comments 21 (consideration in pdf of manuscript): Line 314 Be clear on this names, cause someone related can think that is prednisolone, but for other readers it can be confunsing. Always put the complete name before the given code |
||
|
Response 21: As requested, this point was corrected.
|
||
|
Comments 22 (consideration in pdf of manuscript): Line 324: Male or Female? |
||
|
Response 22: As requested, this point was inserted. It was male mice.
|
||
|
Comments 23 (consideration in pdf of manuscript): Line 325: Why the experiments were not performed with the EESL? |
||
|
Response 23: The following sentence was inserted in the CFA methodology: “In previous work of our research group, mice were subjected to prolonged treatment with ethanolic extract of S. lachnostachys Benth. leaves and this extract exhibited anti-inflammatory parameters in Complete Freund's Adjuvant (CFA) model [8].”
|
||
|
Comments 24 (consideration in pdf of manuscript): Line 344: Just one individual of each sex for the extracts and the isolated compound, but five male and famales of the control? This is a totally misunderstood of the OECD protocol |
||
|
Response 24: We are in accordance with the comments of the editor and reviewer 1 about this test, please verify that this methodology and results were removed in the new version of the manuscript.
|
||
|
Comments 25 (consideration in pdf of manuscript): Line 358: The lack of superscript can generate a complete confusion of the real cell concentration |
||
|
Response 25: This point was corrected in the new version of the manuscript.
|
||
|
Comments 26 (consideration in pdf of manuscript): Line 366: This can be eliminated, once the citation is next to the author`s name |
||
|
Response 26: This point was corrected in the new version of the manuscript.
|
||
|
Comments 27 (consideration in pdf of manuscript): Line 371: Lower case and italic |
||
|
Response 27: These points were corrected in the new version of the manuscript.
|
||
Reviewer 2 Report
Comments and Suggestions for Authors
See the attached review report.

minor grammatical changes needed.
Author Response
|
Response to Reviewer 2 Comments
|
||
|
1. Summary |
|
|
|
Thank you very much for taking the time to review this manuscript. Please find the detailed responses below and the corresponding revisions/corrections highlighted/in track changes in the re-submitted files. |
||
|
2. Point-by-point response to Comments and Suggestions for Authors |
||
|
Comments 1: The authors anti-arthritic potential of the ethanolic extract of S. lachnostachys and fruticuline B. No need to emphasize on photochemical compound as it confuses the readership. The authors could mention this in the introduction. Grammar should be improved. Response: Thanks for pointing this out. However, we think that it is important to emphasize that we have evaluated the anti-inflammatory activity of the isolated compound FRUT B once it is the first time it has been tested in animal models. |
||
|
Comments 2: Abstract a. The authors could shorten the background and methodology and instead give results, including values. Response 2a: As requested, we have rephrased and rewritten the abstract, as you can see in the revised version of the manuscript.
b. Give future perspective using one line. |
||
|
Response 2b: As required, all information’s were inserted.
|
||
|
Comments 3: Introduction a. Give the common name, family, etc. Response 3a: This information was already given in lines 59 and 60.
b. Are there reported therapeutic activities of the FRUT? |
||
|
Response 3b: This information is in the introduction section. |
||
|
Comments 4: Results a. Give NMR spectra in supplementary. Response 4a: There were already NMR data in the Supplementary Material.
b. Data visualization could be improved. Response 4b: Data visualization was improved.
c. Emphasis the difference between Figs. 4 and 5. |
||
|
Response 4c: Both figures give results from carrageenan-induced inflammation modes, however Figure 4 is regarding pleurisy model, and Figure 5 concerns paw edema model.
|
||
|
Comments 5: Discussion a. Inter-relate the results. b. Highlight the relationship between results. c. Compare FRUT with existing drugs. d. How effective this in prevention anti-inflmmatory. ] e. Emphasis on the main conclusion. f. What would be the future perspective?
|
||
|
Response 5: Please note that we inserted the raised points to improve the discussion section.
|
||
|
Comments 6: Methodology a. Who authenticated the plant? Response 6a: Prof. Dr. Élide P. Santos harvested and identified the species. This information was inserted in the new version of the manuscript.
b. What is meant by “three times each solvent 287 at room temperature), EESL and Hex were obtained.” Is it EESL or ethanol extraction? Please clarify. Response 6b: The leaves were extracted three times with the solvents. The dried leaves (415.3 g) were let to be extracted with 2.0 L of hexane for 24 hours, then filtered, and the leaves (filtrated) were extracted again with more 2 L of pure hexane. This procedure was repeated at room temperature three times with hexane and ethanol.
c. Were the authors able to isolate without fractionation process? Response 6c: No. The whole fractionation procedure is explained in section 4.2.
d. What was the negative control used? Response 6d: The saline solution was used as the negative control. For comparison, we used prednisolone as the positive control group.
e. Did the authors include a positive control to compare the effect of the compound? Response 6e: Sure. Prednisolone was the positive control.
f. Give references for methodologies used. Response 6f: We included references for the essays.
g. The authors should follow OECD protocol for toxicity study. Response 6g: We are under the comments of the editor and reviewers about this test; please verify that this methodology and results were removed in the new version of the manuscript.
h. Stastical analysis should be more descriptive. Response 6h: We improved the statistical analysis description. |
||
|
|
||
Round 2
Reviewer 1 Report
Comments and Suggestions for Authors
I have carefully reviewed the revised manuscript, and I am pleased to note that the authors have addressed the necessary modifications as requested. The revisions have significantly improved the clarity and quality of the manuscript, and all concerns raised during the initial review have been adequately addressed.
The methodology and findings are now presented with enhanced precision, and the authors have provided clear and comprehensive responses to my comments. Based on the thorough revisions and the strengthened content, I recommend this manuscript for publication.